# Evapotranspiration Seasonality over Tropical Ecosystems in Mato Grosso, Brazil

**Marcelo Sacardi Biudes** [1], **Hatim M. E. Geli** [2,*], **George Louis Vourlitis** [3], **Nadja Gomes Machado** [4], **Vagner Marques Pavão** [5], **Luiz Octávio Fabrício dos Santos** [5] **and Carlos Alexandre Santos Querino** [6]

1 Instituto de Física, Universidade Federal de Mato Grosso, 2367, Av. Fernando Corrêa da Costa, Cuiaba 78060-900, MT, Brazil; marcelo@fisica.ufmt.br
2 New Mexico Water Resources Research Institute, New Mexico State University, Las Cruces, NM 88003, USA
3 Biological Sciences Department, California State University San Marcos, 333 S. Twin Oaks Valley Rd., San Marcos, CA 92096, USA; georgev@csusm.edu
4 Instituto Federal de Mato Grosso, Av. Juliano da Costa Marques, Cuiaba 78050-560, MT, Brazil; nadja.machado@ifmt.edu.br
5 Programa de Pós-Graduação em Física Ambiental, Instituto de Física, Universidade Federal de Mato Grosso, 2367, Av. Fernando Corrêa da Costa, Cuiaba 78060-900, MT, Brazil; vagner.pavao@rondonopolis.mt.gov.br (V.M.P.); luizoctavio@fisica.ufmt.br (L.O.F.d.S.)
6 Instituto de Educação Agricultura e Ambiente, Universidade Federal do Amazonas, 786, Rua 29 de Agosto, Humaita 69800-000, AM, Brazil; carlosquerino@ufam.edu.br
* Correspondence: hgeli@nmsu.edu; Tel.: +1-575-646-1640

**Abstract:** Brazilian tropical ecosystems in the state of Mato Grosso have experienced significant land use and cover changes during the past few decades due to deforestation and wildfire. These changes can directly affect the mass and energy exchange near the surface and, consequently, evapotranspiration (ET). Characterization of the seasonal patterns of ET can help in understanding how these tropical ecosystems function with a changing climate. The goal of this study was to characterize temporal (seasonal-to-decadal) and spatial patterns in ET over Mato Grosso using remotely sensed products. Ecosystems over areas with limited to no flux towers can be performed using remote sensing products such as NASA's MOD16A2 ET (MOD16 ET). As the accuracy of this product in tropical ecosystems is unknown, a secondary objective of this study was to evaluate the ability of the MOD16 ET ($ET_{MODIS}$) to appropriately represent the spatial and seasonal ET patterns in Mato Grosso, Brazil. Actual ET was measured ($ET_{Measured}$) using eight flux towers, three in the Amazon, three in the Cerrado, and two in the Pantanal of Mato Grosso. In general, the $ET_{MODIS}$ of all sites had no significant difference from $ET_{Measured}$ during all analyzed periods, and $ET_{MODIS}$ had a significant moderate to strong correlation with the $ET_{Measured}$. The spatial variation of ET had some similarity to the climatology of Mato Grosso, with higher ET in the mid to southern parts of Mato Grosso (Cerrado and Pantanal) during the wet period compared to the dry period. The ET in the Amazon had three seasonal patterns, a higher and lower ET in the wet season compared to the dry season, and minimal to insignificant variation in ET during the wet and dry seasons. The wet season ET in Amazon decreased from the first and second decades, but the ET during the wet and dry season increased in Cerrado and Pantanal in the same period. This study highlights the importance of deepening the study of ET in the state of Mato Grosso due to the land cover and climate change.

**Keywords:** MOD16 ET; precipitation; EVI; Amazon; savanna hyper-seasonal; Pantanal

## 1. Introduction

Natural ecosystems in the Brazilian Midwest (Mato Grosso, Goias, Mato Grosso do Sul) play a critical role in controlling local and regional climate [1] by altering the temperature and precipitation patterns [2,3] and partitioning surface energy fluxes [4]. Over the past few decades, these ecosystems have been significantly modified due to anthropogenic and climate change impacts such as deforestation [5,6], degradation, and wildfire [7]. These

impacts directly affect ecosystem functions including the seasonal variability of the mass and energy exchange processes [8]. Evapotranspiration (ET) is a key indicator of these human- and climate change-induced effects over such regionally contrasting agricultural (pasture, soybean, corn, and cotton) and natural ecosystems in the Brazilian Midwest [9,10].

The land cover of the state of Mato Grosso consists of complex vegetation mosaics with a gradient of rainforest, semi-deciduous forest, and shrub and grassy vegetation that can respond differently in terms of water use (i.e., ET) to increased atmospheric temperature and precipitation variability. Northern Mato Grosso is covered mostly with the Amazon biome [Amazon], which is characterized by humid or seasonal tropical forests [4,11]. Central Mato Grosso is covered with the Cerrado biome [Cerrado], which can be considered as the largest savanna in the Americas. The Cerrado has a seasonal climate and xeromorphic vegetation interspersed with grass, herbaceous vegetation, and small woody plants [12]. Southern Mato Grosso is covered with the Pantanal biome [Pantanal], which represents the most extensive floodplain in the world, with topographic variations and the flood pulse (intensity and duration) as moderating forces for landscape diversity and species distribution [13]. In addition to the heterogeneity of natural vegetation, deforestation for agricultural and livestock activities has occurred in Mato Grosso since 1960 [14]. The vegetation mosaic and seasonality of water availability with pronounced dry and wet seasons and a flooding period define the patterns of ET in Mato Grosso [4]. However, these dynamic processes add measurement and modeling challenges in obtaining accurate and representative estimates of ET [4,15,16].

Several methods can provide estimates of ET with comparable accuracies. Ground-based methods can be considered the most accurate ones; however, determining ET by conventional methods is complex, costly, and has spatial limitations [17,18]. Alternatively, determining ET at different spatial and temporal scales can be achieved using remote sensing modeling techniques and products [17,19,20] that allow for developing more feasible and consistent ET datasets such as the Atmosphere–Land Exchange Inverse (ALEXI) [21–23], the operational Simplified Surface Energy Balance (SSEBop) [24,25], and MOD16A2 ET [26,27] [MOD16 ET]. ALEXI is a physically based two-source energy balance model that depends mainly on land surface temperature (LST) as a lower boundary condition. The SSEBop is an empirical model that follows the big leaf concept (assuming the surface can be represented by one homogenous big leaf or an extended green cover), which also depends on LST to estimate ET. MOD16 ET is a physically based model that depends on the Penman–Monteith equation to estimate ET, does not require LST, and it is readily available (since 2000) at 500 m spatial resolution. ET based on ALEXI (4 km) is not currently readily available and is yet to be operational. ET based on SSEBop (1 km) is available only at a monthly time scale for global applications. Moreover, SSEBop uses the big leaf assumption to represent ET over a land surface—an assumption that may not be appropriate to represent the heterogenous surface conditions in this region. As the ET footprint measured in flux towers is smaller than 1 km$^2$ [28], coarse spatial resolution ET products, such as ALEXI and SSEBop, can introduce errors due to losses on surface heterogeneity to compute surface ET [29,30]. Recent evaluations of some of the remote sensing-based estimates of ET in lower latitudes showed variable performance without the indication of the superiority of a specific model to the others over contrasting landscapes [10,31].

The MOD16 ET product is derived from the Moderate Resolution Imaging Spectro-radiometer (MODIS) sensors on the Terra satellites. This product has been widely used to evaluate the space–time dynamics of ET at various spatial scales (e.g., [1]), and was proposed by [26] and modified later by [27]. The MOD16 ET algorithm adapts the Penman–Monteith equation to calculate vegetation transpiration and soil evaporation following a multi-source approach [32,33]. Currently, the 6th version of the MOD16 ET product has modifications made to its ET algorithm to increase the spatial resolution (from 1000 to 500 m), estimate night-time ET, and use an updated land cover product. Other recent MOD16 product updates were described in [33].

MOD16 ET has been validated globally over different ecoregions with data from about 46 flux towers spread across seven types of land cover and in different climates [33,34], including, for example, Savannah in South Africa [35]; bush areas, pasture, and cultivation in Northwest Mexico [36]; irrigated areas in southern Italy [37]; humid regions in China [38]); and an arid area in the United States [39]. However, MOD16 ET was evaluated in only two types of vegetation in Brazil, namely, in natural evergreen broad-leaved forest and in a mixture of small deciduous thorny trees and shrubs in Caatinga [32,34,40]. Unfortunately, these two landforms do not represent the mosaic of Brazilian biomes and ecosystems and, therefore, do not characterize ET seasonal patterns over the wide range of Brazilian tropical ecosystems.

Brazil has six biomes distributed over broad topographic and climatic gradients [41,42], with three biomes (Amazon, Cerrado, and Pantanal) present in Mato Grosso. These biomes represent challenging gradients of climate (temperature and rainfall), land cover (forest, savanna, wetland, and agriculture), and hydrology (upland vs. seasonally flooded), and assessing the satellite-derived ET of these gradients is a stringent test of the MOD 16 ET product. Such a test is needed, given the importance of satellite-based estimates of ET for large, heterogenous tropical environments, and, in particular, to assess the uncertainties in the MOD16 ET product caused by landcover and climate variation [17–19]. Variation in climate, landcover, and hydrology also cause large spatial and temporal variation in ET. For example, within the Amazon biome, ET is reportedly insensitive to seasonal variation in precipitation in the northwestern part of the basin but not the southeastern part of the basin [4,11,15]. In contrast, seasonal variation in precipitation exerts a strong control on ET and energy balance in the Cerrado biome; however, high variability in natural [43] and anthropogenic [3] landcover exerts strong but poorly described spatial and temporal control on ET [4,44,45]. Finally, variation in flood-drying cycles that occur in the Pantanal, partly because of seasonal variation in precipitation and partly because of the hydrological cycles that are decoupled from local precipitation [46], exert strong control on season and interannual ET [4,16]. These climate, landcover, and hydrological variations are profoundly altered by human expansion [1–3,5–8], highlighting the need to evaluate the spatial and temporal variations in ET across a heterogenous and rapidly changing region such as the state of Mato Grosso.

The goal of this study was therefore to characterize the seasonal and decadal patterns of ET over these biomes that are critical to understand vegetation response to climate change and to support studies related to Mato Grosso in the future. To achieve this goal, a consistent long-term remote sensing, ground-based observation, or a combination of ET information was needed. Thus, we compiled existing ground-based and remotely sensed ET datasets to: (1) characterize the long-term temporal variability in ET derived from ground-based measurements and the MOD16 ET product; (2) characterize the uncertainties associated with the MOD 16 ET; (3) assess relationships between regional ET and temporal (seasonal, annual, and decadal) trends in precipitation, air temperature, and vegetation productivity. This analysis was conducted to complement several other studies (e.g., [4,11,15,16,44,45]) that were aimed to examine and better understand seasonal, interannual, and long-term climate effects on natural ecosystems in Mato Grosso.

## 2. Materials and Methods

### 2.1. The Study Sites

The study was conducted over the entire state of Mato Grosso, Brazil, using eight representative experimental sites (three in Amazon, three in Cerrado, and two in Pantanal) with different soil water and vegetation dynamics, distributed across the state in the north-south direction (Figures 1 and 2). Field measurements were acquired during different periods over each site between 2001 and 2020 (Table 1). The first site (referred to herein as **AFL**) was in a dense, evergreen ombrophilous forest located 39 km northeast of the city of Alta Floresta in the southern Amazon Basin (9°36′2.83″S: 55°55′22.22″W). The vegetation is composed of *Tetragastris altissima* (22%), *Celtis schippii* (17%), and *Pseudolmedia* sp. (6%),



with an average canopy height of 30–35 m but with some emergent trees reaching up to 45 m in height [4]. The 30-year mean annual temperature in the Alta Floresta region is 25.7 °C, and precipitation is approximately 2230 mm year$^{-1}$, with a dry season from June to September [47]. The soil is classified as a Ultisol, and it is acidic (pH = 4.5), with low levels in phosphorus, extractable cation, and organic matter content [48]. The second site (referred to herein as **SIN**) is an Amazon–Cerrado transition forest located in the northern region of the state of Mato Grosso, 50 km from the city of Sinop (11°24′45″S: 55°19′30″W, 423 m above of the sea). The vegetation in SIN is composed mainly of several species that include *Brosimum lactescens*, *Qualea paraensis*, and *Tovomita schomburkii*. The canopy height over SIN ranges between 25 and 28 m, and the Leaf Area Index (LAI) varied between 4.8 m$^2$ m$^{-2}$ during the wet season and 4.2 m$^2$ m$^{-2}$ during the dry season [49]. The soil is an Entisol (Quartzipsamments), characterized by approximately 90% sand, low pH (4.2), and low fertility. The average air temperature in the region is about 24.7 °C, with an annual precipitation of approximately 2000 mm year$^{-1}$ that falls during the rainy season followed by a dry season of 5 months (May–September) [4]. The third site (referred to herein as **FSN**) is a pasture located in the Fazenda São Nicolau near Cotriguaçu (9°51′43.8″S, 58°13′48.6″W). The site at FSN is dominated by a non-native *Brachiaria brizantha* pasture, and the soil was classified as Oxisol [50].

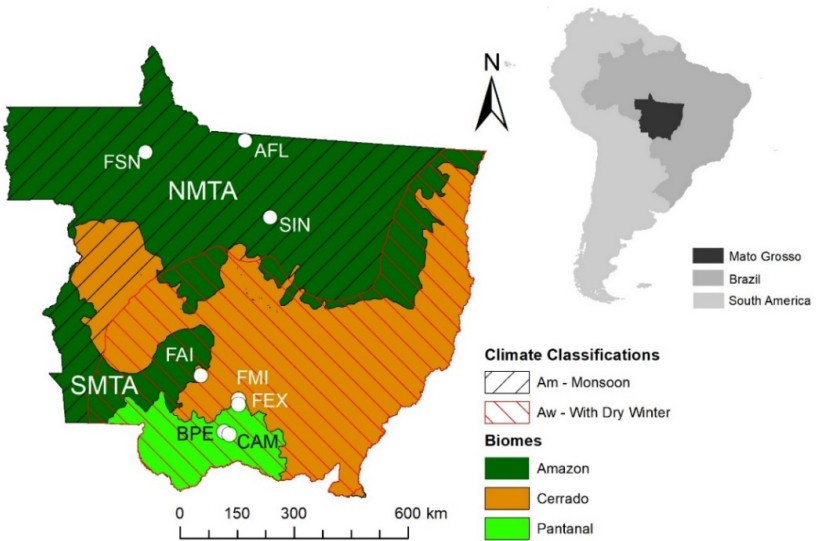

**Figure 1.** Location of flux towers in the evergreen ombrophilous forest (AFL), Amazon–Cerrado transition tropical forest (SIN), *Brachiaria brizantha* pasture (FSN) in the Amazon, Stricto Sensu Cerrado (FAI), mixed woodland–grassland (FMI), *Brachiaria humidicola* pasture (FEX) in the Cerrado, floodplain areas of Cambará monodominant forest (CAM) and shrubs (BPE) in the Pantanal and the Köppen's climate classification [41] in the State of Mato Grosso, Brazil. The Amazon biome in this study was identified as North Mato Grosso Amazon (NMTA) and South Mato Grosso Amazon (SMTA).

The fourth site (referred to herein as **FAI**) is in the *Stricto Sensu* Cerrado and is located at Fazenda Arco-Iris, 20 km of Barra do Bugres (15°10′38.88″S, 56°58′3.41″W) (Figure 1). The region's soil is Entisol [51], with high rates of water infiltration but with reduced water retention capacity [52]. The fifth site (referred to herein as **FMI**) is in the Cerrado, and is located at Fazenda Miranda, 15 km south of Cuiabá (15°43′53.66″S: 56°04′18.81″W) (Figure 1). The vegetation at the FMI is characterized by a mixed woodland–grassland (locally known as campo sujo) dominated by native and non-native grasses along with semi-deciduous tree species *Curatella americana* L. and *Diospyros hispida* A.DC. [43]. The region's soil is Oxisol [51], with high rates of water infiltration but with a reduced water retention capacity [52]. The average monthly air temperature ranges from 18 °C in June–July to a maximum of 29 °C in October, and the average annual precipitation is 1420 mm year$^{-1}$, followed by a dry season from May to September [53]. Nearby, the sixth site was in

a non-native grassland located at the Fazenda Experiment (referred to herein as **FEX**), 33 km south of Cuiabá (15°51′15.23″S: 56°04′13.50″W). The pasture at FEX is dominated by the non-native *Brachiaria humidicola* [4]. The regional soil type at both research sites is rocky, dystrophic Oxisol [4,51], with high rates of water infiltration but with a limited water-holding capacity [4].

The seventh and eighth sites are in a hyper-seasonal flooded region in the Pantanal (Figure 1). The third site (referred to herein as **BPE**) is located at an experimental site called Baía das Pedras, 105 km to the southwest of Cuiabá (16°29′53.52″S: 56°24′46.23″W). The vegetation in BPE is predominantly the *Combretum laxum* type, known locally as "Pombeiral", which is a large shrub up to 3 m in height that grows in very dense thickets throughout the Pantanal [16]. The fourth site (referred to here as **CAM**) is an area of monodominant vegetation from *Vochysia divergens* Pohl (Cambará), known locally as "Cambarazal", with a canopy height that varies between 28 and 30 m, with an average LAI of 3.5 m$^2$ m$^{-2}$ located at Reserva Particular do Patrimônio (RPPN) SESC Pantanal, 107 km distant from Cuiabá (16°33′19.11″S: 56°17′11.49″W) [49]. The topography of both experimental sites (i.e., BPE and CAM) is flat, which favors the formation of water depths between 1 and 2 m during the wet season [54], and the soil is classified as Entisol. The average annual temperature in the region is 26.1 °C and has an average precipitation of 1400 mm during the rainy season followed by a dry season from May to September [4,49].

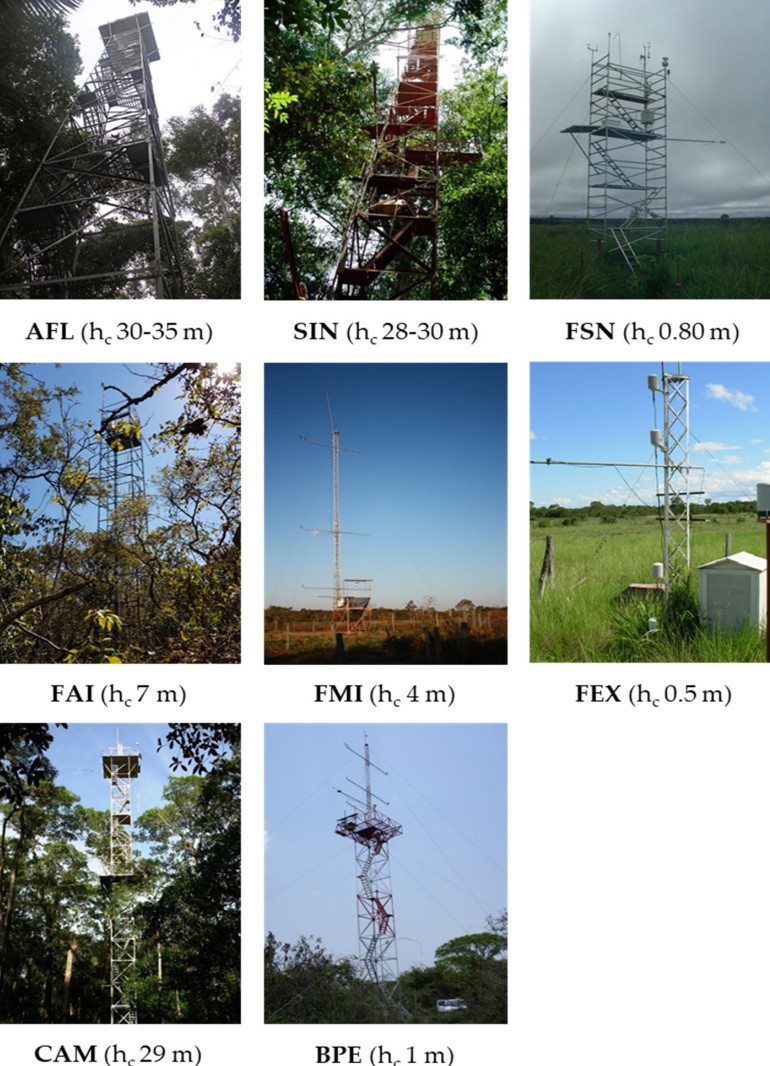

**Figure 2.** The Flux towers and the canopy height (h$_c$) at the eight study sites AFL, SIN, FSN, FAI, FMI, FEX, CAM, and BPE.

**Table 1.** A summary of the field measurements periods over the study sites AFL, SIN, FSN, FAI, FMI, FEX, CAM, and BPE.

| | Biome | Site | Description | Data Availability |
|---|---|---|---|---|
| **1** | | **AFL** | dense, evergreen ombrophilous forest near the city of Alta Floresta | 2002–2003 |
| **2** | **Amazon** | **SIN** | Amazon–Cerrado transition forest near Sinop City | 2001–2008 |
| **3** | | **FSN** | non-native *Brachiaria brizantha* pasture in the Fazenda São Nicolau near Cotriguaçu | 2002–2003 |
| **4** | | **FAI** | Cerrado *Stricto Sensu* located at Fazenda Arco-Iris near Barra do Bugres | 2019–2020 |
| **5** | **Cerrado** | **FMI** | mixed woodland–grassland at Fazenda Miranda near Cuiabá city | 2009–2013 |
| **6** | | **FEX** | non-native grassland located at the Fazenda Experimental near Cuiabá | 2006–2009 |
| **7** | **Pantanal** | **BPE** | seasonal flooded large shrubs at Baía das Pedras near Cuiabá city | 2010–2015 |
| **9** | | **CAM** | Monodominant seasonal flooded forest from Cambará at RPPN SESC Pantanal near Cuiabá city | 2006–2009 |

### 2.2. Precipitation and EVI

The time series of precipitation data for each study site were obtained from the Global Precipitation Measurements (GPMs) of the Giovanni Platform of the National Aeronautics and Space Administration (NASA) (https://giovanni.gsfc.nasa.gov/giovanni/, accessed on 27 September 2021). The enhanced vegetation index (EVI) is a vegetation index that represents vegetation growth conditions and is based on the 16-day 250 m Terra Moderate Resolution Imaging Spectroradiometer (MODIS) Vegetation Indices (MOD13Q1) version 6, based on the geo-location information (latitude and longitude) of each experimental area at the Google Earth Engine platform. The EVI can provide indications about vegetation greenness and cover (leaf area) over time in response to available energy for photosynthesis and water supply. The EVI is well suited over areas with dense vegetation and high biomass compared to other indices such as the Normalized Difference Vegetation Index (NDVI). Vegetation indices such as EVI are usually used to estimate canopy biophysical properties such as LAI and height—both variables are used in different ET models.

### 2.3. MOD16 ET Product

Estimates of ET based on the MOD16A2 version 6 product (referred to herein as MOD16 ET) were obtained and processed on the Google Earth Engine platform. MOD16 ET is available at a spatial resolution of 500 m at 8-day temporal resolution [35]. The pixels from MOD16 ET that coincided with each flux tower location were used to conduct the comparison between the ground-based measurements and remotely sensed estimates. The MOD16 ET product includes a quality assurance (QA) band which is designed to help users understand the data. We only used data with high QA in this study.

MOD16 ET, which is based on the Penman–Monteith equation [34,55], is executed daily, as it accounts for daytime and night-time ET [27]. In general, MOD16 ET is the sum of soil evaporation, wet canopy evaporation, and canopy transpiration. MOD16 ET product uses two groups of input data. The first group of data is based on the Global Modeling and Assimilation Office (GMAO/MERRA) that provides daily meteorological reanalysis, which includes solar radiation, air temperature, air pressure and humidity, current night vapor pressure, night air temperature, and incident shortwave radiation [33]. The second group consists of products derived from remote sensing, which include surface albedo based on MCD43A3, the fraction of photosynthetically active radiation (FPAR), and the LAI of MOD15A2H and land cover classes based on MODIS land cover type 3 product (MCDLCHKM).

One of the main factors to be considered in the MOD16 ET estimates is the biome property lookup table (BPLUT). This table consists of the constants of physical and biophysical parameters which vary according to the type of classified surface. The logic used in MOD16 ET estimates assumes that the specific physiological parameters of each biome do not change for different species of a given biome (for example, savana and its different

structural variations in Brazil) or at any time during the year [33]. The detailed BPLUT formulation, calibration, and validation process can be found in [27,56].

### 2.4. ET Measurements

Surface energy balance and micrometeorological measurement were acquired at eight towers (Figure 2). Each flux tower was equipped with instruments to measure net radiation (Rn), air temperature (Tair), relative humidity (RH), and soil heat flux (G). All eight towers had similar and consistent sensors, data acquisition, as well as recording and storage systems. The sensors used include an NRLITE (Kipp & Zonen, Delft, The Netherlands) to measure Rn, HMP-45AC (Vaisala Inc., Woburn, MA, USA) to measure Tair/RH and an HFP01 (Hukseflux BV, Delft, The Netherlands) to measure G as well as a CR1000 datalogger (Campbell Scientific, Inc., Logan, UT, USA) to read and store the data.

Mean $ET$ (ET; mm 30-min$^{-1}$) for each flux tower was calculated using the Bowen ratio (BWR) method (Equation (1)). It was assumed that the gradients were sufficient in each experimental area due to the relative homogeneity of the terrain. The criteria used in the selection of data calculated by the BWR method were those described by Perez et al. [57] and reviewed by Hu et al. [58]. Daily ET was obtained by adding the 48 ET values of those averaged at 30-min intervals. Daily $ET$ was integrated every eight days (referred to herein as $ET_{Measured}$; mm 8-day$^{-1}$) so that it could be compared with the 8-day MOD16 ET (referred to herein as $ET_{MODIS}$; mm 8-day$^{-1}$).

$$ET = \left( \frac{LE}{\lambda} \right) \tag{1}$$

where $LE$ is the latent heat flux (W m$^{-2}$) (Equation (2)) and $\lambda$ is the latent heat of vaporization (Equation (3)).

$$LE = \frac{Rn - G - \Delta S}{1 + \beta} \tag{2}$$

$$\lambda = 1.919.10^{-6} \left( \frac{T + 273.16}{(T + 273.16) - 33.91} \right)^2. \tag{3}$$

where $Rn$ is the net radiation (W m$^{-2}$), $G$ is the soil heat flux (W m$^{-2}$), $\Delta S$ is the storage heat flux(W m$^{-2}$) in the canopy and the biomass calculated following the parameterization proposed by Moore and Fisch [59], and $T$ is the mean air temperature (°C). $\Delta S$ was calculated every 30 min in AFL, SIN, and CAM; however, $\Delta S$ was neglected in the other sites due to the low density of vegetation. The Bowen ratio, $\beta$, was calculated as:

$$\beta = \left( \frac{Cp}{0.622\lambda} \right) \left( \frac{\Delta T}{\Delta e} \right) \tag{4}$$

where $Cp$ is the specific heat at constant pressure (1.00467 J g$^{-1}$ K$^{-1}$), 0.622 is the proportion of molecular weights of water and air, and $\Delta T$ and $\Delta e$ are the difference in air temperature (°C) and water vapor pressure (kPa) between the two measurement levels, respectively.

More details about the calculation of ET using the Bowen ratio method used in this study, as well as on filling in gaps caused by sensor failure and/or data rejection, were according to the criteria established by Perez et al. [57], and those particularly for these sites can be found in Biudes et al. [4].

### 2.5. Statistical Analysis

The annual, seasonal, and monthly averages with their respective confidence intervals ($\pm 95\%$) of $ET_{Measured}$ and $ET_{MODIS}$ were calculated using the bootstrapping resampling technique with 1000 interactions. Annual refers to calculating the average of all 8-day ET data points for the entire period of the study over each site. The wet and dry seasons corresponded to October–April, May–September, and the calendar year, respectively. In order to highlight the long-term seasonal amplitude in $ET_{MODIS}$ over the three biomes, the

averages of the two driest months (i.e., August–September) and the three wettest months (i.e., February–April) during the decades of 2000–2009 and 2010–2019 were selected.

The accuracy of the $ET_{MODIS}$ product was seasonally assessed using the Willmott ($d$) agreement index proposed by Willmott et al. [60] (Equation (5)), Mean Absolute Error ($MAE$) (Equation (6)), and Root Mean Square Error ($RMSE$) (Equation (7)).

$$d = 1 - \left[ \frac{\sum (P_i - O_i)^2}{\sum (|P_i - O| + |O_i - O|)^2} \right] \tag{5}$$

$$MAE = \sum \frac{|P_i - O_i|}{n} \tag{6}$$

$$RMSE = \sqrt{\frac{\sum |P_i - O_i|^2}{n}} \tag{7}$$

where $P_i$ are the estimated values, $O_i$ are the observed values, $O$ is the average of the observed values, and $n$ is the number of observations. The Willmott agreement index (referred to herein as Willmott coefficient) presents the degree of performance of an estimate based on the distance between the estimated and observed values. The Willmott coefficient ranges from 0 (no agreement) to 1 (perfect arrangement) [61]. The MAE indicates the absolute average (deviation) distance. The RMSE indicates the model's failure to estimate the variability of measurements around the mean and measures the variation of the estimated values around those measured. The lower limit of RMSE is 0, which represents a perfect agreement between the model's estimates and the measured data.

## 3. Results

### 3.1. Comparison of $ET_{Measured}$ and $ET_{MODIS}$

Regionally, estimates of ET based on $ET_{MODIS}$ showed not statistical differences when compared with measurements, $ET_{Measured}$, during all analyzed periods (i.e., annual, dry, and wet) with combined data from all sites (referred to as ALL in Figure 3). There was a moderate correlation (r = 0.51, *p*-value < 0.001) between $ET_{MODIS}$ and $ET_{Measured}$, as the Wilmott agreement index (d) was approximately 0.68 (see ALL in Table 2; Figure 4). However, the performance of $ET_{MODIS}$ varied over the different regions (i.e., AMZ, CER, and PAN). In the Amazon (i.e., AMZ in Figure 3), $ET_{MODIS}$ consistently overestimated $ET_{Measured}$ during all analyzed periods by about 38–43%, while it underestimated those in the Cerrado (i.e., CER in Figure 3) by about 43% during the dry season (Figure 3). $ET_{MODIS}$ in Cerrado during the annual and wet periods was not statistically different from $ET_{Measured}$. Similarly, $ET_{MODIS}$ was not statistically different compared with $ET_{Measured}$ in the Pantanal (i.e., PAN in Figure 3) in all analyzed periods. There was a strong correlation (r = 0.71, *p*-value < 0.001) between $ET_{MODIS}$ and $ET_{Measured}$ in the Cerrado, while it was moderate in the Amazon (r = 0.54, *p*-value < 0.001) and Pantanal (r = 0.61, *p*-value < 0.001) (Table 2). The highest d of 0.77, and lowest MAE of 6.3 mm 8-day$^{-1}$ and RMSE of 8.2 mm 8-day$^{-1}$, occurred in the Cerrado, while the lowest d of 0.44, and the highest MAE of 9.5 mm 8-day$^{-1}$ and RMSE of 10.7 mm 8-day$^{-1}$ occurred in the Amazon.

Locally, the performance of $ET_{MODIS}$ over the individual sites also showed some variability. $ET_{MODIS}$ consistently overestimated $ET_{Measured}$ over all Amazon sites (i.e., AFL, SIN, and FSN) during all analyzed periods (Figure 3). The lowest d of 0.32, and the highest MAE of 13.7 mm 8-day$^{-1}$ and RMSE of 14.6 mm 8-day$^{-1}$, occurred in FSN, while an inverse pattern occurred in AFL (Table 2; Figure 4). A strong correlation (r = 0.79, *p*-value < 0.001) between $ET_{MODIS}$ and $ET_{Measured}$ was shown in FSN, while it was moderate in AFL (r =0.55, *p*-value < 0.001) and SIN (r = 0.62, *p*-value < 0.001). Over the Cerrado sites (i.e., FAI, FMI, and FEX), $ET_{MODIS}$ and $ET_{Measured}$ were not statistically different, except during the dry season in FAI and FEX, over which $ET_{MODIS}$ underestimated $ET_{Measured}$ by about 50% and 52%, respectively (Figure 3). The values of the Willmott $d$ were greater than 0.73, and those of r were higher than 0.68 over the Cerrado sites (Table 2; Figure 4). Over the Pantanal

sites (i.e., CAM and BPE), $ET_{MODIS}$ and $ET_{Measured}$ were not statistically different during all periods (Figure 3); moreover, there was a moderate correlation (r > 0.60), and the Willmott *d* < 0.65 (Table 2; Figure 4).

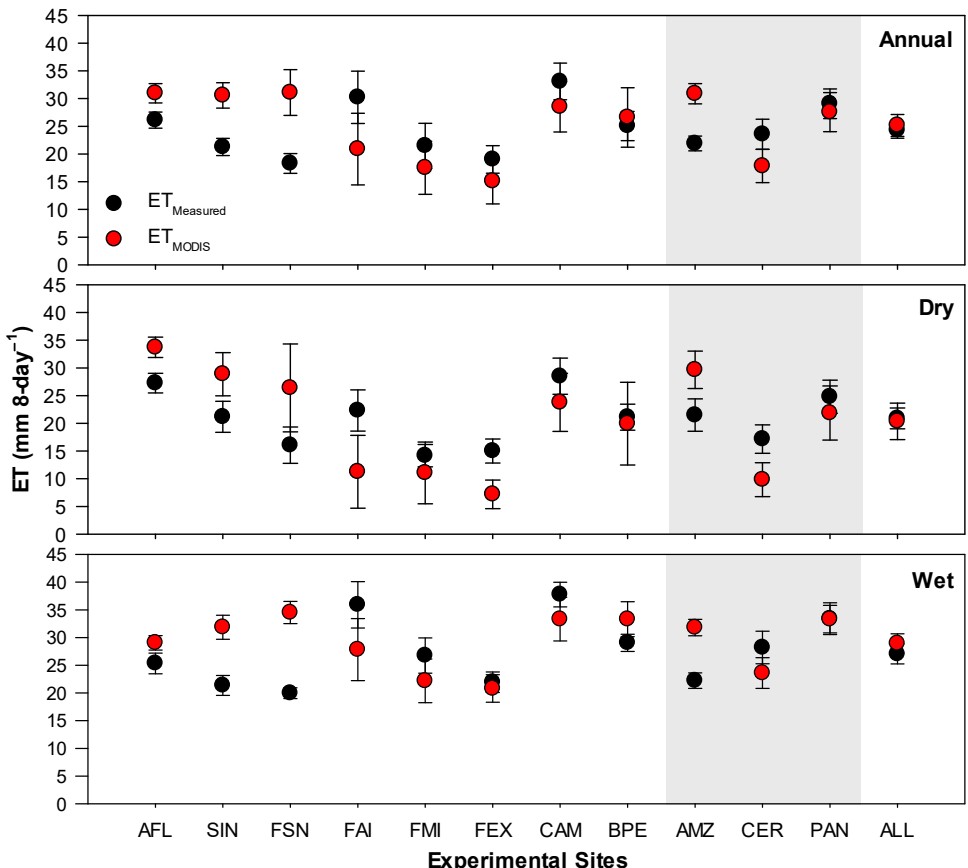

**Figure 3.** Annual and seasonal mean (±95% Confidence Interval) of measured ($ET_{Measured}$) and estimated ($ET_{MODIS}$) 8-day evapotranspiration (mm 8-day$^{-1}$) in each site in the Amazon (AMZ), Cerrado (CER) and Pantanal (PAN) sites and in all sites (ALL). The shading area refers to the mean ET in the three biomes of Mato Grosso.

**Table 2.** Summary of Pearson's correlation coefficient (r), Willmott agreement index (d), Mean Absolute Error (MAE; mm 8-day$^{-1}$), and Root Mean Square Error (RMSE; mm 8-day$^{-1}$) of the measured and estimated evapotranspiration (i.e., $ET_{Measured}$ and $ET_{MODIS}$, respectively) accumulated every 8 days (mm 8-day$^{-1}$) for the entire period of the study in each site in the Amazon (AMZ), Cerrado (CER), and Pantanal (PAN) sites and all sites collectively (ALL). Values accompanied by (***) indicate *p*-value < 0.001.

|  | Site | *r* | *d* | MAE (mm 8-day$^{-1}$) | RMSE (mm 8-day$^{-1}$) |
|---|---|---|---|---|---|
| **Amazon** | **AFL** | 0.55 *** | 0.49 | 5.1 | 5.5 |
|  | **SIN** | 0.62 *** | 0.45 | 9.4 | 10.5 |
|  | **FSN** | 0.79 *** | 0.32 | 13.7 | 14.6 |
| **Cerrado** | **FAI** | 0.89 *** | 0.74 | 10.9 | 11.7 |
|  | **FMI** | 0.68 *** | 0.78 | 5.4 | 7.5 |
|  | **FEX** | 0.72 *** | 0.73 | 5.2 | 6.6 |
| **Pantanal** | **CAM** | 0.60 *** | 0.65 | 7.4 | 9.6 |
|  | **BPE** | 0.71 *** | 0.73 | 6.9 | 7.9 |
| **Amazon (AMZ)** |  | 0.54 *** | 0.44 | 9.5 | 10.7 |
| **Cerrado (CER)** |  | 0.71 *** | 0.77 | 6.3 | 8.2 |
| **Panatanal (PAN)** |  | 0.61 *** | 0.73 | 7.1 | 8.7 |
| **ALL** |  | 0.51 *** | 0.68 | 7.7 | 9.4 |

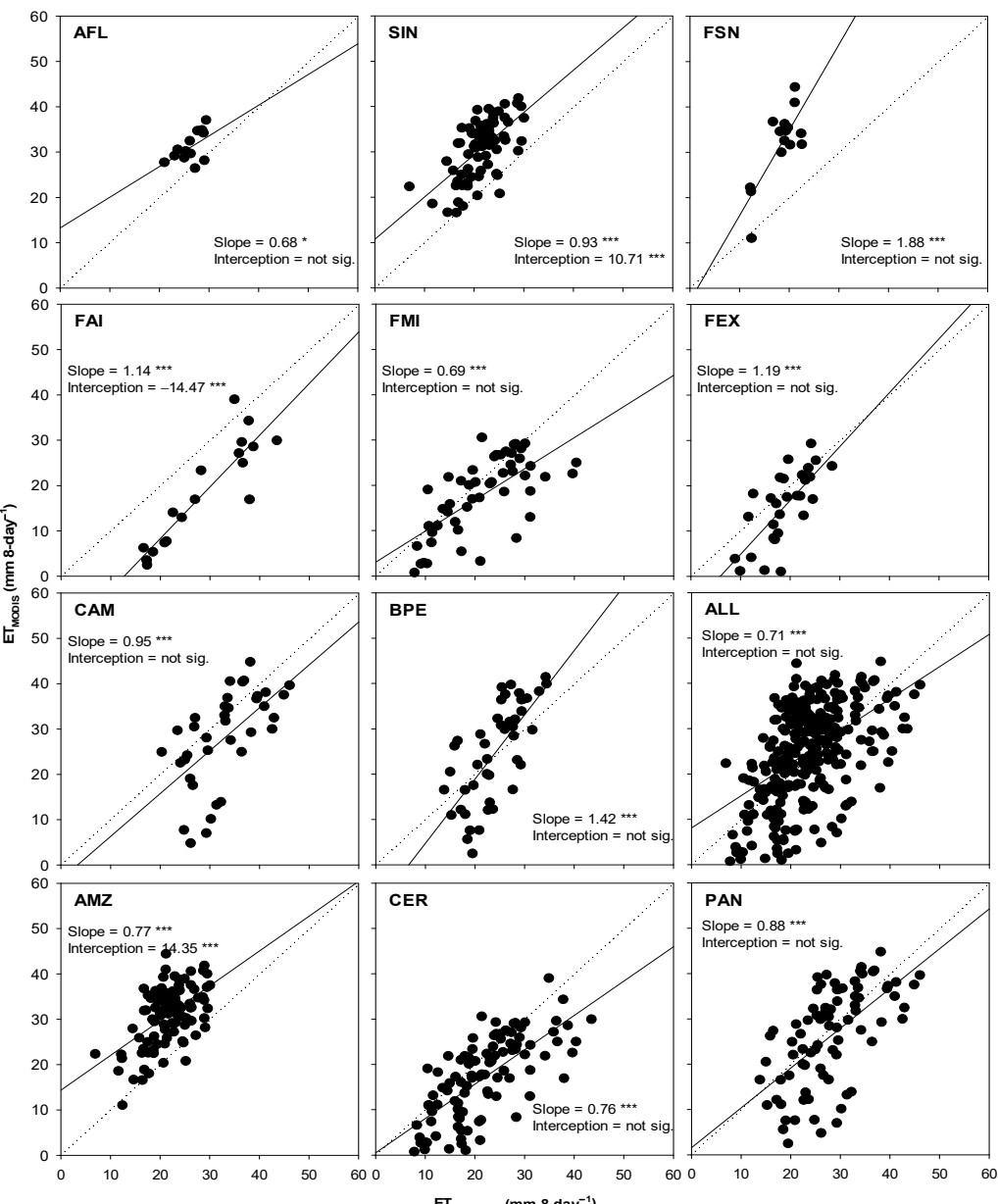

**Figure 4.** Scatterplot of cumulative measured (ET$_{Measured}$) and estimated (ET$_{MODIS}$) 8-day evapotranspiration (mm 8-day$^{-1}$) in each site in the Amazon (AMZ), Cerrado (CER), and Pantanal (PAN) sites, and then in all sites (ALL). The slope and intercept coefficients values of the regression are accompanied by (*) indicate *p*-value < 0.05, and (***) *p*-value < 0.001.

### 3.2. Temporal Patterns in ET, Precipitation, and EVI

The total annual precipitation over the Amazon (i.e., averaged over AFL, SIN, and FSN) of 2077 mm year$^{-1}$ was higher than that over the Cerrado of 1632 mm year$^{-1}$ (i.e., averaged over FAI, FMI, and FEX) and Pantanal of 1525 mm year$^{-1}$ (i.e., averaged over CAM and BPE). Around 90% of the annual precipitation was recorded during the wet season (October–April) (Table 3). The monthly precipitation for all sites usually reaches a peak value during the period between December and March and decreases around May, which marks the beginning of a dry season. In this study, the dry season was marked as the period when the total monthly precipitation was <100 mm [62]—which usually occurs between April and September. It was noticed that the duration of the dry season was about 4–5 months in the Amazon sites and 5–6 months in the Cerrado and Pantanal sites (Figure 5).

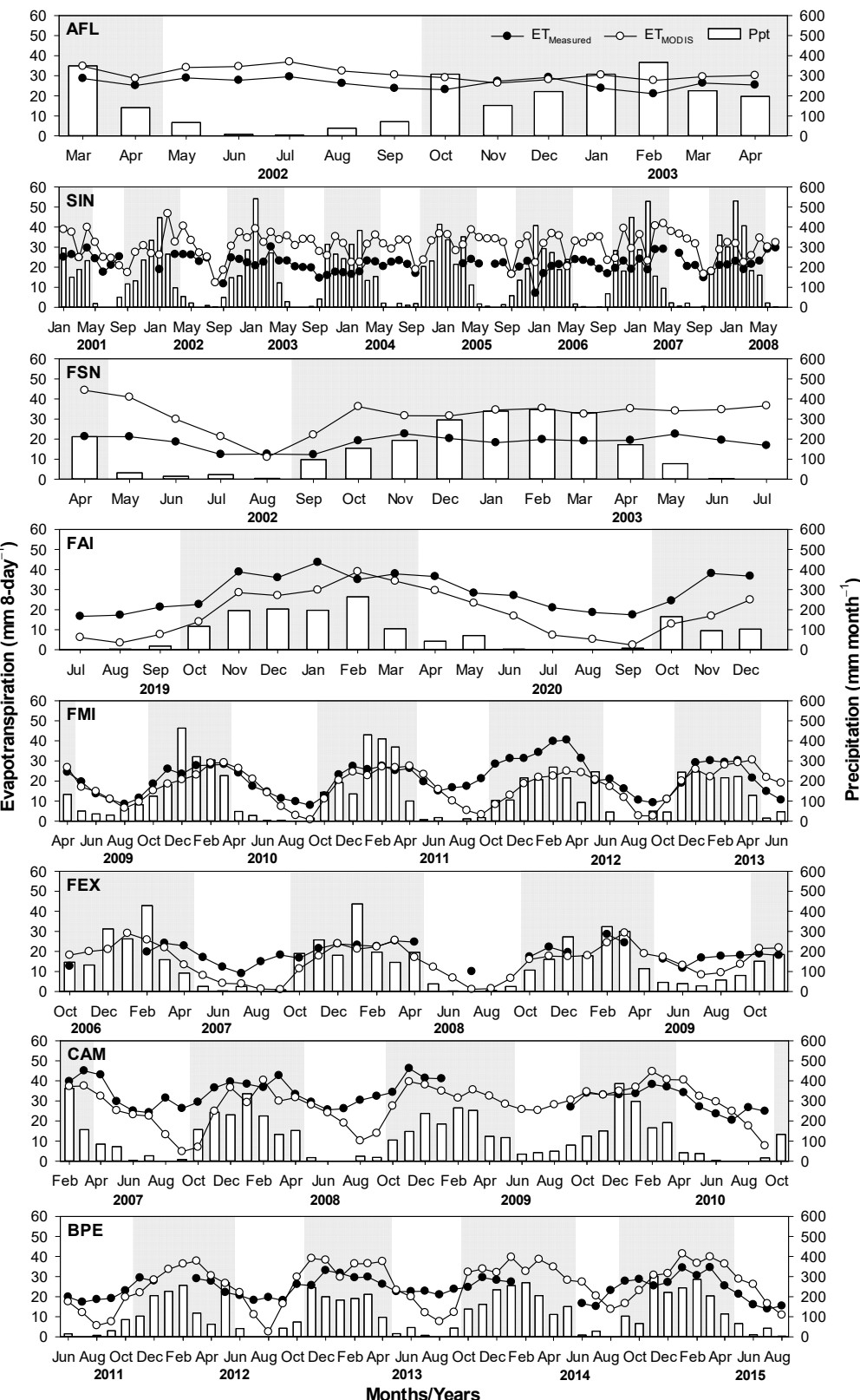

**Figure 5.** Monthly cumulative precipitation (Ppt) and monthly average of measured (ET$_{Measured}$) (black circles) and estimated (ET$_{MODIS}$) (white circles) evapotranspiration of cumulative 8 days (mm 8-day$^{-1}$), respectively, in the experimental sites located in the Amazon (AFL, SIN and FSN), Cerrado (FAI, FMI and FEX), and Pantanal (CAM and BPE). The wet seasons over each site were highlighted with a gray shaded background. A threshold of more than 100 mm of precipitation was used to identify the dry season based on [62].

**Table 3.** Annual and seasonal accumulated precipitation (Ppt; mm) and mean (95% confidence interval) of the enhanced vegetation index (EVI; dimensionless) in each site, in the Amazon (AMZ), Cerrado (CER) and Pantanal (PAN) sites, and all sites (ALL).

| Site | | Precipitation (Ppt) (mm) | | | EVI | | |
|---|---|---|---|---|---|---|---|
| | | Annual | Dry (May–Sept.) | Wet (Oct.–April) | Annual | Dry (May–Sept.) | Wet (Oct.–April) |
| Amazon | AFL | 2198 | 191 | 2007 | $0.54 \pm 0.03$ | $0.54 \pm 0.02$ | $0.54 \pm 0.05$ |
| | SIN | 1896 | 102 | 1794 | $0.50 \pm 0.02$ | 0.51 0.01 | $0.49 \pm 0.02$ |
| | FSN | 2138 | 184 | 1953 | $0.44 \pm 0.04$ | $0.42 \pm 0.05$ | $0.45 \pm 0.05$ |
| Cerrado | FAI | 1625 | 154 | 1471 | $0.37 \pm 0.04$ | $0.29 \pm 0.04$ | $0.42 \pm 0.03$ |
| | FMI | 1643 | 145 | 1498 | $0.33 \pm 0.03$ | $0.27 \pm 0.02$ | $0.37 \pm 0.02$ |
| | FEX | 1628 | 144 | 1484 | $0.37 \pm 0.03$ | $0.30 \pm 0.03$ | $0.42 \pm 0.01$ |
| Pantanal | CAM | 1530 | 256 | 1274 | $0.50 \pm 0.03$ | $0.46 \pm 0.02$ | $0.54 \pm 0.04$ |
| | BPE | 1521 | 251 | 1270 | $0.45 \pm 0.04$ | $0.39 \pm 0.04$ | $0.52 \pm 0.04$ |
| Amazon (AMZ) | | 2077 | 159 | 1918 | $0.49 \pm 0.02$ | $0.49 \pm 0.03$ | $0.49 \pm 0.03$ |
| Cerrado (CER) | | 1632 | 148 | 1485 | $0.36 \pm 0.02$ | $0.29 \pm 0.02$ | $0.40 \pm 0.02$ |
| Pantanal (PAN) | | 1525 | 254 | 1272 | $0.48 \pm 0.03$ | $0.43 \pm 0.03$ | $0.53 \pm 0.03$ |
| All sites (ALL) | | 1745 | 187 | 1558 | $0.44 \pm 0.02$ | $0.40 \pm 0.03$ | $0.47 \pm 0.02$ |

The enhanced vegetation index (EVI) had relatively higher values in the Amazon ($0.49 \pm 0.02$) and Pantanal ($0.48 \pm 0.03$) compared to those in the Cerrado ($0.36 \pm 0.02$) (Table 3). The seasonal variation of vegetation greenness based on the EVI was similar to that of precipitation in the Cerrado and Pantanal sites (r = 0.52–0.76; Table 4), with relatively high values during the wet season and low values during the dry season (Table 3). However, the EVI of the Amazon sites showed minimal to no seasonal variability, i.e., it indicated no significant correlation with precipitation (Tables 3 and 4). The annual EVI was higher in AFL and lower in FMI. The highest seasonal variability in EVI occurred in FAI, FMI, FEX, and BPE, and the lowest occurred in CAM (Figure 5; Table 3). EVI is generally used to monitor vegetation growth conditions; thus, it was able to depict the seasonal variation of the canopy greenness that was closely responsive to precipitation. The EVI showed higher values during the wet season in the Cerrado and Pantanal following the same temporal behavior of precipitation. The minimum EVI varied between the sites, with the lowest values occurring during the middle of the wet season in the Amazon sites and later in the dry season in the Cerrado and Pantanal. The highest EVI values were observed from October to December in all sites (Figure 6).

**Table 4.** Pearson's correlation coefficient r between cumulative measured ($ET_{Measured}$) and estimated ($ET_{MODIS}$) 8-day evapotranspiration (mm 8-day$^{-1}$), precipitation (Ppt; mm), and enhanced vegetation index (EVI; dimensionless) in the experimental sites. Values accompanied by (*) indicate *p*-value < 0.05, (**) *p*-value < 0.01, and (***) *p*-value < 0.001.

| Region | Amazon | | | Cerrado | | | Pantanal | |
|---|---|---|---|---|---|---|---|---|
| Sites | AFL | SIN | FSN | FAI | FMI | FEX | CAM | BPE |
| **Variables** | | | | | | | | |
| $ET_{Measured} \times Ppt$ | −0.47 | −0.21 | 0.37 | 0.70 ** | 0.62 *** | 0.64 *** | 0.63 *** | 0.72 *** |
| $ET_{Measured} \times EVI$ | −0.03 | 0.20 | 0.43 | 0.93 *** | 0.80 *** | 0.58 *** | 0.55 *** | 0.82 *** |
| $ET_{MODIS} \times Ppt$ | −0.47 | 0.02 | 0.31 | 0.80 *** | 0.60 *** | 0.79 *** | 0.56 *** | 0.77 *** |
| $ET_{MODIS} \times EVI$ | −0.29 | 0.03 | 0.57 * | 0.89 *** | 0.73 *** | 0.88 *** | 0.38 * | 0.80 *** |
| $Ppt \times EVI$ | −0.29 | −0.27 ** | 0.05 | 0.76 *** | 0.72 *** | 0.67 *** | 0.52 *** | 0.74 *** |

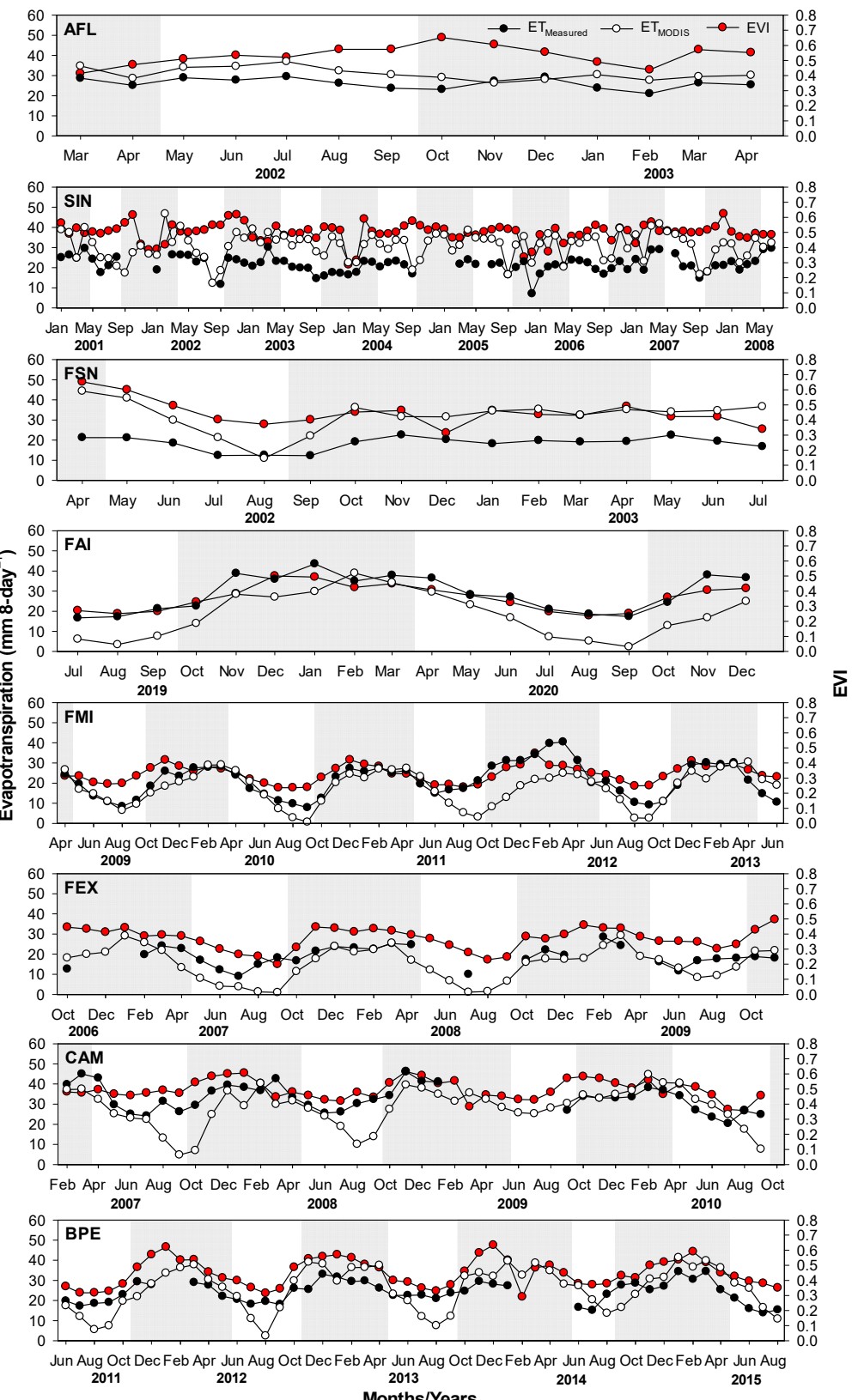

**Figure 6.** Monthly average of measured (ET_Measured) (black circles) and estimated (ET_MODIS) (white circles) evapotranspiration of cumulative 8 days (mm 8-day$^{-1}$), respectively, and 16-day average MODIS-based enhanced vegetation index (EVI) (red circles) in the experimental sites located in the Amazon (AFL, SIN and FSN), Cerrado (FAI, FMI and FEX), and Pantanal (CAM and BPE). The wet season over each site was highlighted with a gray vertical shaded background. A threshold of more than 100 mm of the monthly precipitation was used to identify the dry season, based on [62].

Both $ET_{Measured}$ and $ET_{MODIS}$ had similar patterns with positive correlation with precipitation and EVI over the Cerrado and Pantanal sites, but with insignificant correlation over the Amazon sites (Table 4). Both $ET_{Measured}$ and $ET_{MODIS}$ in the Cerrado and Pantanal sites showed consistent seasonal amplitude, while those ET over the Amazon sites had a gentle amplitude. The highest values of $ET_{Measured}$ occurred over the Pantanal during all analyzed periods, and the lowest ones occurred over the Cerrado and Amazon (Figure 3). However, $ET_{MODIS}$ indicated a different pattern, as it showed higher values over the Amazon, followed by the Pantanal and Cerrado. Moreover, both $ET_{Measured}$ and $ET_{MODIS}$ showed decreased values at the beginning of the dry season at all sites; however, $ET_{MODIS}$, at the Cerrado and Pantanal sites, showed a sharper decrease than $ET_{Measured}$ from July onwards, and diverged considerably from $ET_{Measured}$ in September, when its values were close to zero (Figures 5 and 6). In general, the highest $ET_{Measured}$ values occurred during January to March, while the highest $ET_{MODIS}$ values occurred mostly from November to March (Figures 5 and 6).

*3.3. Long-Term Spatiotemporal Trends in ET*

The spatial variability of seasonal averaged $ET_{MODIS}$ over the last two decades (2000–2009 and 2010–2019) is shown in Figure 7. The average of $ET_{MODIS}$ was taken for two different seasons that included the dry season from May to September and the wet season from October to April. During the dry season in the 2000–2009 decade, $ET_{MODIS}$ was clearly higher in the northern parts of Mato Grosso compared to the southern parts of the state. The northern parts, where the AFL, SIN, and FSN were located (Figure 7a) in the Amazon biome, is referred to, hereafter, as the Northern Mato Grosso Amazon (NMTA) (Figure 1). The southern parts of Mato Grosso, where FAI, FMI, and FEX and CAM and BPE were located, is the region that is mostly covered with the Cerrado and Pantanal biomes. The $ET_{MODIS}$ in the NMTA region during the dry season was mostly > 25 mm 8-day$^{-1}$. However, the part of the Amazon that lies within southwest Mato Grosso (SMTA) (Figure 1) showed much lower $ET_{MODIS}$ with values mostly below 15 mm 8-day$^{-1}$ during the dry season. On the other hand, both the Cerrado and Pantanal showed $ET_{MODIS}$ values mostly below 10 mm 8-day$^{-1}$ during this season. During the wettest months (Figure 7b), $ET_{MODIS}$ in NMTA was predominantly higher than 35 mm 8-day$^{-1}$ in both decades, while the rest of Mato Grosso had values that ranged from 10 to 40 mm 8-day$^{-1}$ during the first decade (2000–2009), and higher than 25 mm 8-day$^{-1}$ during the second decade (2010–2019). This general spatial pattern of higher ET in north Mato Grosso and lower ET in south Mato Grosso was a characteristic of both dry and wet seasons and was also consistent during the past two decades, albeit with some local variations, as explained below.

The amplitude and long-term trend in ET over Mato Grosso were assessed based on the differences in ET between wet and dry seasons over the past two decades and are presented in Figures 8 and 9. The difference (or amplitude) in the average $ET_{MODIS}$ between the wet (October–April) and dry (May–September) seasons for each decade was estimated as a percent of the driest period, as shown in Figure 8. The spatiotemporal pattern in $ET_{MODIS}$ appeared in terms of consistently higher ET values during the wet season compared to the dry season in the southern parts of the state. However, over the northern parts of the state, the ET showed mixed patterns, with higher values during the wet season in some areas—an opposite pattern of higher values during the dry season in the most northern parts in the first decade and in the northwest part in the second decade—and minimum/insignificant variation in ET over relatively small areas. One of the patterns indicated that the amplitude in $ET_{MODIS}$ in SIN, AFL, and FSN and their surrounding areas was moderate (20–40% Figure 8a), with higher ET in the wet season compared to the dry season during the first decade. However, the $ET_{MODIS}$ in these sites showed minimal or opposite seasonality during the second decade, with higher ET during the dry season compared to the wet season. The seasonal amplitude in the SMTA showed about 40–60% increase in ET during the wet season compared to the dry season. In the eastern parts of Cerrado and most of the Pantanal, the amplitude was particularly pronounced, with a

40–80% increase in ET in the wet season compared to that of the dry season during the first decade and 20–40% during the second decade.

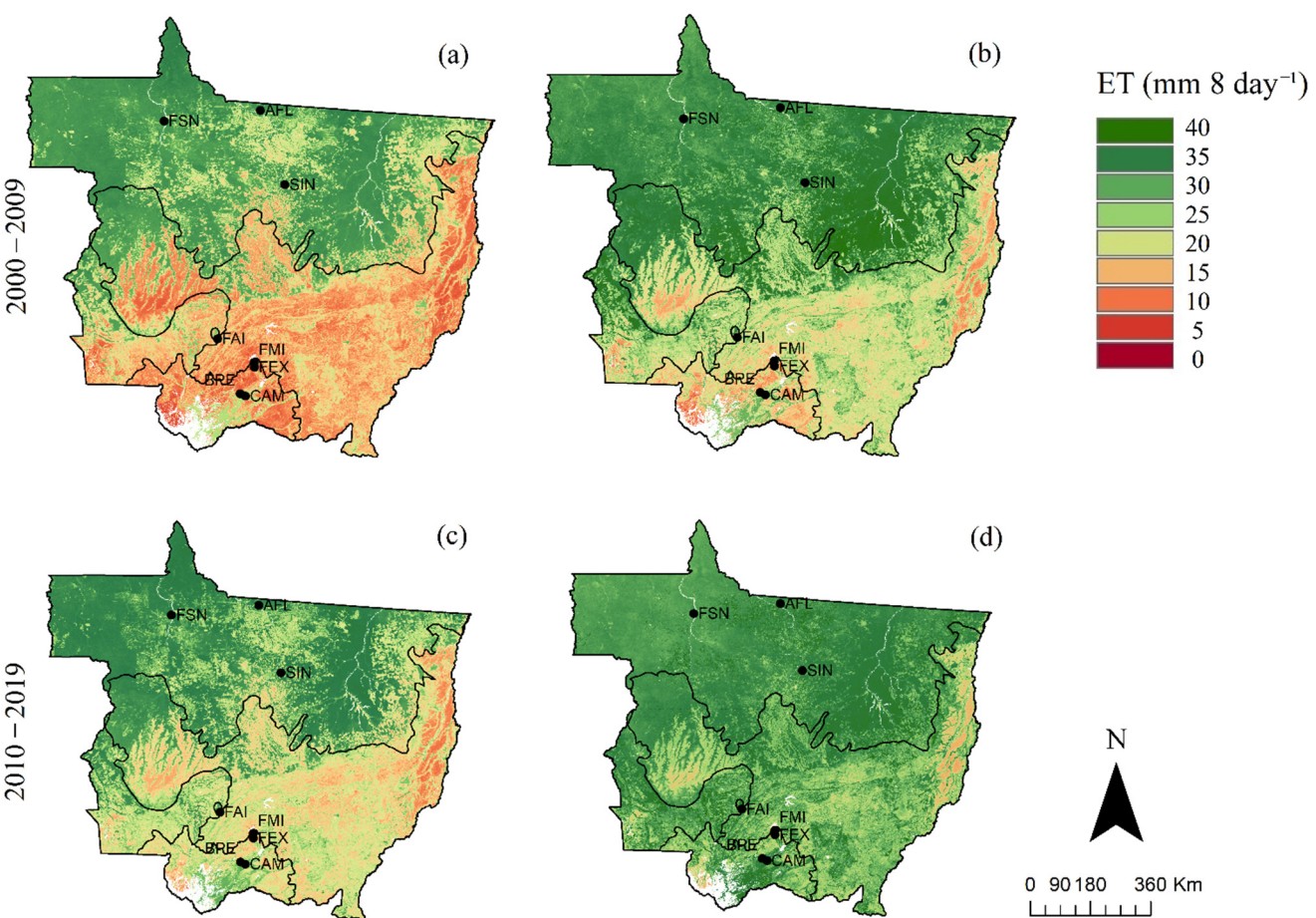

**Figure 7.** Averaged ET$_{MODIS}$ during the dry (May–September; panels (**a**,**c**)) and wet seasons (October–April; panels (**b**,**d**)) in 2000–2009 (panels (**a**,**b**)) and 2010–2019 (panels (**c**,**d**)).

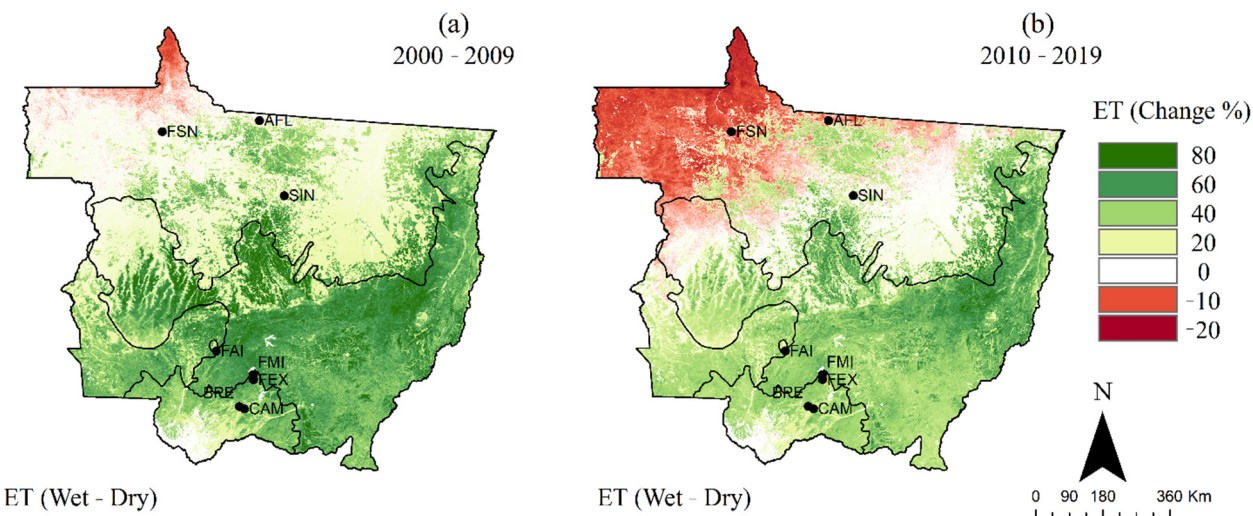

**Figure 8.** The relative difference (amplitude) in average ET$_{MODIS}$ in the state of Mato Grosso from the dry (May–September) to wet (October–April) seasons during the (**a**) 2000–2009 and (**b**) 2010–2019 decades. Positive difference indicates that the average ET$_{MODIS}$ in the wet season is higher than that in the dry season.

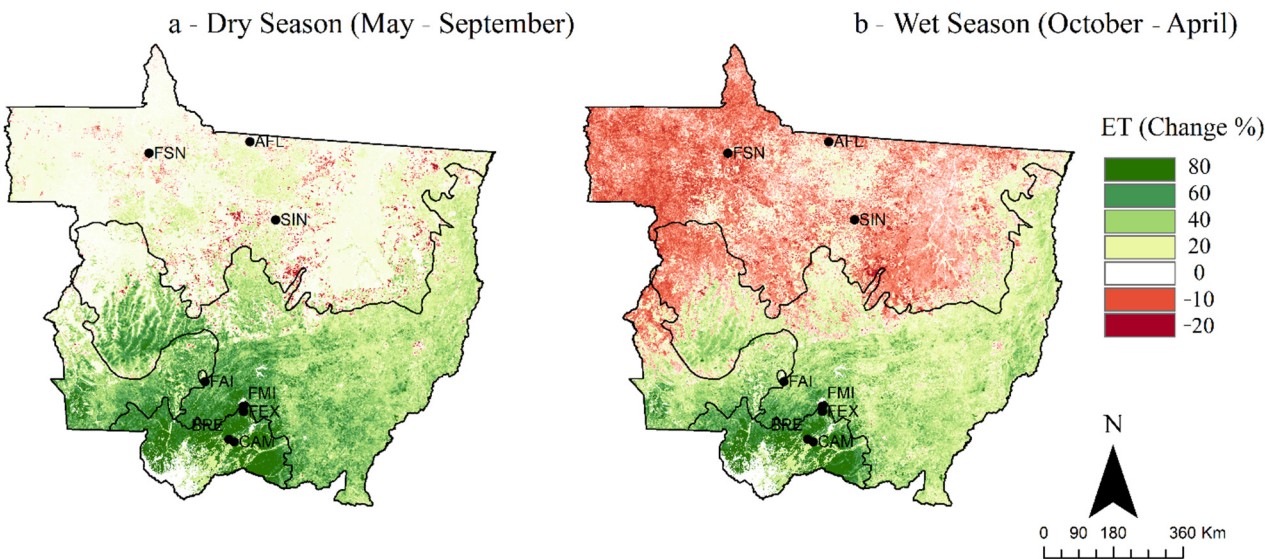

**Figure 9.** Relative difference in the average ET$_{MODIS}$ of the state of Mato Grosso between 2000–2009 and 2010–2019 during the (**a**) dry season (May–September) and (**b**) wet season (October–April). Positive relative difference indicates that the average ET$_{MODIS}$ in 2010–2019 is higher than in 2000–2009.

The long-term trend in ET over Mato Grosso was evaluated by taking the difference of the average ET$_{MODIS}$ of the dry season for the 2010–2019 and 2000–2009 decades (Figure 9a). A similar analysis was conducted for the ET during the wet season (Figure 9b). The results indicated that an increase in ET during the dry season in the past decade (i.e., 2010–2019), compared to that of the 2000–2009 decade, occurred mainly in southern Mato Grosso, with some regions showing a decrease. The results also indicated a decrease in ET during the wet season in the 2010–2019 decade compared to that of the 2000–2009 decade that occurred mostly in northern Mato Grosso (Figure 9b). The ET values during the dry season over most parts of NMTA increased by about 0–20%; however, some small areas of NMTA exhibited a decrease in ET of about 0–20% in 2010–2019 compared to 2000–2009 (Figure 9a). The Cerrado, Pantanal (mainly the southern parts), and SMTA showed an increase greater than 60% in ET$_{MODIS}$ during the dry season in the 2010–2019 decade compared to those of the 2000–2009 period. On the other hand, the ET$_{MODIS}$ decreased by around 10% in the northern half of the state in the wet season during the 2010–2019 decade compared to that of the 2000–2009 decade (Figure 9b).

## 4. Discussion

### 4.1. ET$_{MODIS}$ Performance

The combined ET$_{MODIS}$ dataset of all sites indicated that the MOD16 ET product was not significantly different from the measured ET during all analyzed periods; however, there were some notable spatial and temporal discrepancies between measured and MOD16 ET. This variable performance can be partly attributed to the different patterns of soil–plant–atmosphere interactions and the MOD16 ET model parametrization. Previous studies indicated that ET in the Cerrado sites (FAI, FMI, and FEX) was limited by soil water availability [4,44], while the ET in the evergreen ombrophilous forest (AFL) and the seasonally flooded forest (CAM) was limited by solar radiation availability [4]. The ET in the deciduous forest (SIN), the pasture (FSN), and the seasonally flooded shrubs (BPE) was limited by both factors—i.e., soil water and solar radiation availability [11,15,16,44,50,63].

The complexity of the energy and mass exchange processes in the studied sites can influence the accuracy of the MOD16 ET product. For example, the semi-deciduous forest in SIN decreases the number of leaves during the first part of the dry season [49], as indicated by the observed EVI; nonetheless, ET remains high during this season [4,15]. One of the main reasons for this behavior can be attributed to the fact that the maximum

stomatal conductance (saturated light) in SIN in the upper canopy (20 and 28 m high) and the medium canopy (12 m high) respond in different amplitudes to seasonality [11]. The understory of the seasonally flooded forest in CAM loses leaves, while there is leaf expansion at the overstory in the forest during the dry season [49]. Still, CAM has a ratio of latent heat flux to net radiation (i.e., LE/Rn) that is practically constant (or with minimal variation) throughout the year, even in the dry season, as indicated in [4]. The canopy in SIN and CAM site have well-defined overstory and understory portions (i.e., extracts from middle to top portion of canopy, and middle to ground portion of canopy, respectively) and respond differently to environmental conditions throughout the year [11,64]. Thus, the different performance challenges of the MOD16 ET product over these two sites, for example, can be partly related to the lack of sensitivity (or ability) of the MODIS sensor as well as to the MOD16 ET algorithm to obtain the biophysical properties of the forest and its understory in SIN, and thus in modeling ET over such heterogeneous environments.

The MOD16 ET product tended to overestimate ET in the Amazon sites (AFL, SIN, and FSN) and underestimate ET in the Cerrado during the dry season (Figure 3). This behavior can be attributed to a lack of precision of the MOD16 ET algorithm to estimate canopy and soil conductance. Canopy transpiration is controlled by the canopy conductance, which, in turn, represents the average stomatal conductance status at the leaf level. Thus, changes in the canopy conductance directly modify the canopy's transpiration and, consequently, ET [26,27]. Estimates of MOD16 ET in the Cerrado and Pantanal sites (and to some extent in SIN and FSN) in September at the end of the dry season reached values close to 0 mm 8-day$^{-1}$, coincidentally when a high vapor pressure deficit (VPD) and no precipitation occurred [4], which highlights a limitation of the MOD16 ET algorithm in estimating ET during the months with the greatest water deficit on the soil surface and in the atmosphere.

The vapor pressure deficit (VPD) is one of the main variables used to estimate the stomatal conductance in MOD16 ET [33]. The canopy conductance in MOD16 ET [65] is calculated by multiplying a maximum canopy conductance by a number of scalars that are functions of VPD among other variables. The scalers range from 0 to 1 as, the higher the VPD, the lower the scalar and the canopy conductance. Generally, high VPD (so as the evaporative demand) can result in increased ET when the soil water content in the root zone is not limited [66]. However, the combination of high VPD and limited root zone water content can cause the stoma to close and inhibit the transpiration process [66]. In all experimental sites, the VPD increases during the dry season (especially in September), as indicated in [4]. High VPD has different effects on ET over each site. The vegetation in the Cerrado sites and BPE in the Pantanal have a superficial root system that limits their ability to absorb water from deeper soil layers, while the trees in AFL, SIN, and CAM have a deep root system that allows to absorb water from deeper soil layers during the dry season [4,11]. This noticeable decrease in the estimated ET by the MOD16 product was not observed in measured values in September, which shows that the MOD16 ET overestimated the stomatal resistance due to the increase in VPD in the Cerrado and Pantanal sites, based on research findings over FMI [44,45]. Furthermore, the MOD16 ET algorithm calculates ET as the sum of daytime and night-time ET. However, MOD16 ET assumes complete closure of stomata overnight, contradicting research findings by [38]. With this assumption, MOD16 ET consequently underestimates ET, particularly over the Cerrado, since 1 + 3 to 28% of the daily ET occurs at night in response to increased daytime VPD, especially during the dry season [67].

Another potential source of error in MOD16 ET is related to its built-in land cover classification and the biome-specific physiological parameters described in its Biome Property Look Up Table (BPLUT). The BPLUT lists canopy resistance values related to water availability, air temperature, and biome-specific biophysical properties according to the land cover classification [32], assuming that these values do not vary over space and time, and that they are also the same for the different species of a biome. Moreover, the BPLUT is based on the MODIS 500-m land cover classification, which can introduce some classification errors, especially over areas with heterogenous cover types [27].

Except for showing some challenges during the end of the dry season, the MOD16 ET performance, overall, was consistent and within the level of accepted accuracies [36,68–73]. Although correlation and agreement coefficients and errors vary from site to site, the moderate to strong significant correlation between ET values, estimated by the MOD16 product, and the errors obtained in each site, were within the range of 0.27–0.82 and 6.12–21.81 mm 8-day$^{-1}$, respectively, obtained by other studies [35,38,74]. Thus, it was reasonable to use MOD16 ET for the remainder of this analysis with caution, considering the spatial and temporal variations in its accuracy and that the limitations of MOD16 ET should not hinder it use. The variable performance of MOD16 ET within Mato Grosso is in agreement with the findings that no single remote-sensing ET model that can perform well over all biomes, climate, and environmental conditions currently exists, as indicated in [71,72,75].

### 4.2. Biome Responses to Precipitation as Indicated by EVI and ET

The relatively lower ET$_{Measured}$ in the FSN, FMI, and FEX (Figure 3) compared to the other sites can be partially attributed to the type of vegetation, soil, and climate [4]. The vegetation at FSN and FEX were composed of grasses and FMI was composed of grasses interspersed with small tree-shrubs; both have low LAI and, consequently, low transpiration [45]. The soil type of these three sites has high porosity and a low water-holding capacity, which quickly drains water within 3–5 days after a rain event [4,44,50,64], thus limiting root zone water availability for vegetation. The effects of this combination of short, sparse, and shallow-rooted vegetation and low water-holding capacity were reflected by the pattern of the observed EVI.

On the other hand, the hyper-seasonal flooded (between January and June) monodom-inant forest (CAM) has enough water available in the soil during the wet season (i.e., based on the soil type and its water-holding capacity [63,76,77]), and vegetation with a deep root system, to maintain relatively high rates of transpiration during the dry season [4,49]. Similarly, the ombrophilous forest (AFL) and semideciduous forest (SIN) have tall, leafy canopies and a deep root system for absorbing water in the deep-water table even during drought [11,78]. These characteristics have, consequently, resulted in the lowest seasonal amplitude of the measured ET in AFL and SIN when compared to the other sites. The hyper-seasonal flooded shrubland (BPE) has relatively similar hydrological conditions as CAM but has a different ET response due to differences in vegetation cover type. While their vegetation types are different, both BPE and CAM have relatively similar vegetation greenness response in the months before the flood as indicated by the EVI. However, their response diverges during and after the flood period. The EVI over the BPE site decreases from the beginning of the flood (or by the peak of the rainy season in December–January) until August, while, over the CAM site, the canopy leaves are usually well maintained during the flood period [4,49]. The vegetation in BPE has a less deep root system than in CAM [79] and is therefore more susceptible to two extremes of water-related stress situations imposed by excess water during flooding and low soil water availability during the dry season [4].

### 4.3. Long-Term Trend and Spatial Variability in ET in Mato Grosso

The identified spatial variability of ET based on the MOD16 product over Mato Grosso during the dry and wet seasons in this study showed some consistency with the regional climatology that was described in some previous studies (e.g., [4,44,49,80]) as well as the vegetation composition [81]. The spatial pattern of ET shown in Figure 6 was similar to that of precipitation, which had higher and lower values in the northern and southern parts of the state, respectively [82,83].

The amplitude in ET in all sites (except those in the Amazon) indicated higher ET during the wet seasons compared to the dry seasons, as all sites were in regions that exhibit relatively similar seasonal characteristics similar to the Cerrado and Pantanal [4,44]. However, the ET in the Amazon experienced three different seasonal patterns, i.e., higher

and lower ET in the wet season compared to the dry season and minimal to no significant variation in ET during the wet and dry seasons, which have also been reported by previous studies (e.g., [80,84,85]).

The spatial and seasonal patterns of ET in Mato Grosso is the combined result of the effect of climate distribution and type of vegetation. The climate of the Brazilian Midwest, where the state of Mato Grosso is located, is controlled by tropical and subtropical large and mesoscale systems [86]. The local convection, Bolivian High, and South Atlantic Convergence Zone (SACZ) are the main systems responsible for precipitation variability and extreme events in the wet season [87]. SACZ is a system with a northwest–southeast orientation from the Amazon to the subtropics, which passes through Mato Grosso [88]. The high-level anticyclone centered over Bolivia, the "Bolivian High", is generated from intense convective heating, which causes convection and, consequently, latent heat release from the surface to the atmosphere during the summer months in the Amazon [89]. The drought in the Brazilian Midwest is caused by the formation of an anticyclone circulation— a region of high pressure at middle and low levels of the atmosphere—which causes a downward air movement, leading to precipitation suppression and long periods of drought [87]; in addition, cold fronts ("friagens") cause low temperatures at mid-latitudes during May due to polar air in the region [90]. The highest ET values in the northern parts of Mato Grosso coincide with the intersection of the Amazon and the Am Koppen's climate classification [41]. The vegetation in this region is dense and tall, and has a deep root system that enables plants to absorb water from the deep-water table, even during the dry season [78]. Thus, the heating of the surface of the Amazon rainforest causes the local convection of humidity [91].

Changes in ET over time (i.e., 2000–2009 vs. 2010–2019) also varied between the three different regions. For the Amazon, dry season ET remained largely unchanged between 2000–2009 and 2010–2019 (Figure 9a); however, there was a 10–20% decrease in wet season ET for the Amazon during the same period (Figure 9b). In contrast, the Cerrado exhibited a modest increase (20–40%) in ET during the wet and dry seasons, and the Pantanal region exhibited even larger increases in the wet and dry seasons between 2000–2009 and 2010–2019 (Figure 9a,b).

These variations in ET coincided with long-term trends in precipitation and temperature change that were observed over the last four decades (Figure 10). For example, most of the state experienced declines in dry season precipitation over the last 40 years (Figure 10a); however, during the wet season, the northwest part of the state experienced an increase, the NE and middle part of the state experienced no change, and the extreme eastern and southern part of the state experience in a decline in precipitation (Figure 10b). At the same time, dry season temperatures increased in the northern part of the state, especially in the extreme northwest and, to a lesser extent, in the mid- and southern part of the state (Figure 10c). Thus, the dry season became drier and warmer over the last four decades, especially for the extreme northwest part of the state. Temperature trends were also positive for most of the state during the wet season, especially for the extreme eastern part of the Cerrado belt (Figure 10d). Thus, for the Amazon, the extreme northwest part became warmer and wetter while the rest of the Amazon became warmer, with no appreciable change in wet season precipitation. The Cerrado also became warmer during the wet season, however the eastern fringe also became drier, and the Pantanal became both warmer and drier over the last four decades.

How did these changes in climate affect changes in ET? It seems that the warmer and drier conditions during the dry season had little to no effect on Amazon ET. This might be due to the deeper rooting depth of Amazon trees, which provide access to deeper water sources [4,15,84]. However, both warming and wetting (northwest Amazon) and the warming alone (rest of the Amazon) coincided with a decline in ET over the last two decades. Perhaps an increase in cloud-cover reduced ET for the extreme northwest Amazon [92]; however, it is unclear why a decline in wet season rainfall would cause a reduction in ET during the wet season but not the dry season. The increase in Cerrado

ET during the wet and dry season is even more perplexing given the relatively strong warming and drying observed for both seasons. ET for Cerrado has been shown to decline drastically during the dry season [44,45], suggesting that long-term trends in warming and drying should also cause ET to decline. Finally, the strongest increase in ET between 2000–2009 and 2010–2019 was observed in the Pantanal, where presumably high surface water availability due to high water tables and/or standing water from seasonal flooding caused a higher rate of evaporation as the climate became warmer and drier [84,85].

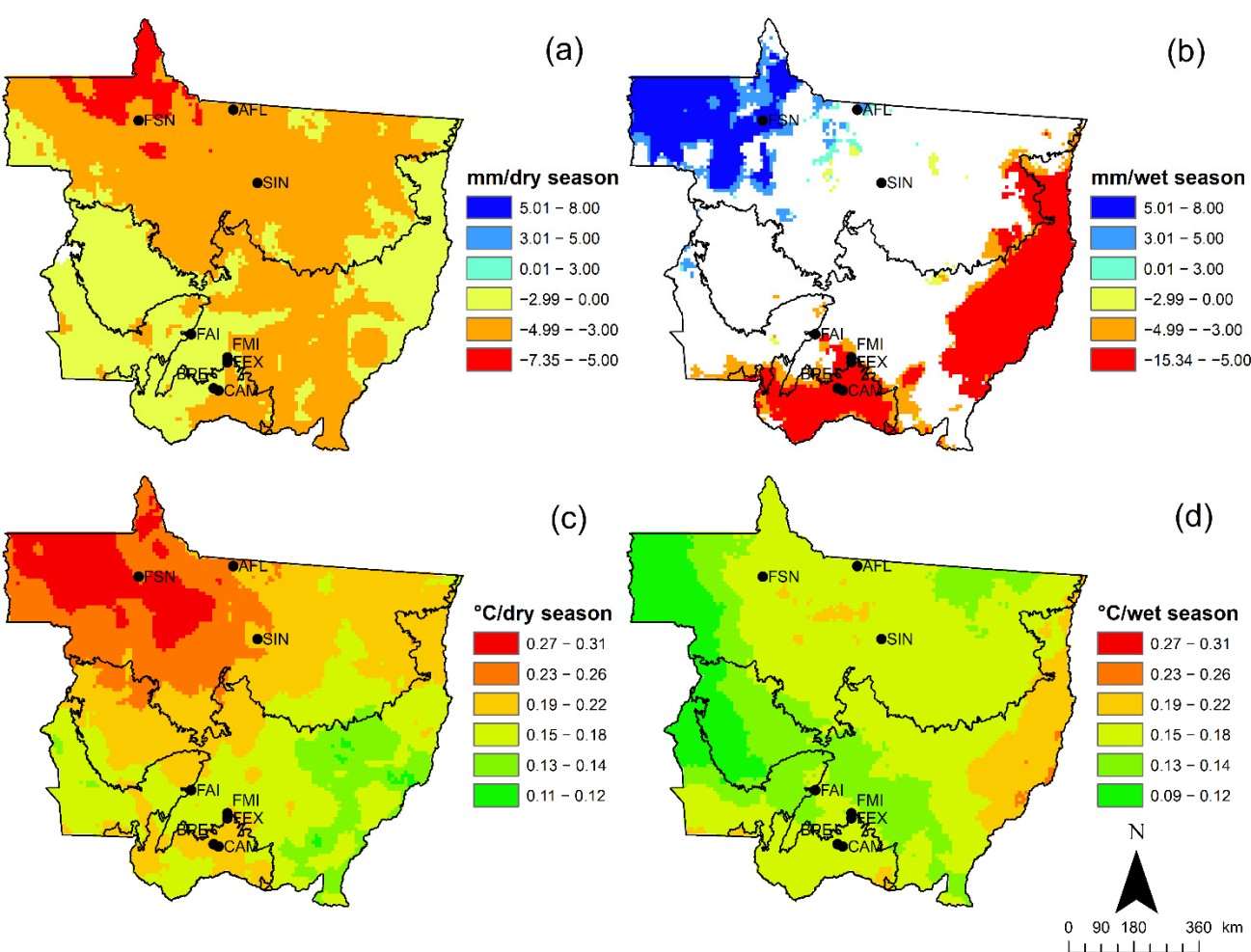

**Figure 10.** ERA-5 precipitation trend per dry (**a**) and wet (**b**) season and temperature trend per dry (**c**) and wet (**d**) season for the last 40 years for the state of Mato Grosso. White space means non-significant trends.

## 5. Conclusions

The goal of this study was to characterize the spatial and seasonal patterns of ET over Mato Grosso tropical ecosystems. The highest values of measured ET occurred in the Pantanal, and lower ones occurred in the Amazon and Cerrado. A clear seasonal amplitude in ET was indicated by remote-sensing and ground-based observations over the five sites, i.e., FAI, FMI, FEX, CAM, and BPE, while ALF, SIN, and FSN showed minimal to no seasonal amplitude.

The ET of all sites estimated by the MOD16 product had no significant difference from measured ET during all analyzed periods. However, the MOD16 product consistently overestimated ET in the Amazon sites and underestimated ET during the months with the greatest vapor pressure deficit. Nevertheless, the ET estimated by the MOD16 product had a significant moderate to strong correlation with the measured ET, and the errors

were within the range determined in other studies, which demonstrates that the use of the MOD16 product is reasonable for the analysis of the spatial and temporal variation of ET.

The spatial variation of ET estimated by the MOD16 product had some similarity to the climatology of Mato Grosso, with higher ET in the mid to southern parts of Mato Grosso (Cerrado and Pantanal) during the wet period compared to the dry period. However, the Amazon experienced three seasonal patterns in ET, i.e., higher and lower ET in the wet season compared to the dry season, and minimal to no significant variation in ET during the wet and dry seasons. The wet season ET in the Amazon decreased over the 2000–2009 decade to the 2010–2019 decade, however the ET during the wet and dry season increased in the Cerrado and Pantanal between 2000–2009 and 2010–2019.

The results on the validation of the MOD16 ET product and on the spatial and seasonal pattern of ET in the state of Mato Grosso can support studies on changes in agroecological zoning, water footprint, and sustainable water consumption in irrigated plantations. In addition, this study highlights the importance of deepening the study of ET in the state of Mato Grosso due to the land cover and climate change.

**Author Contributions:** Conceptualization, M.S.B., N.G.M. and H.M.E.G.; methodology, M.S.B., N.G.M. and C.A.S.Q.; software, V.M.P., L.O.F.d.S. and M.S.B.; validation, M.S.B., V.M.P., N.G.M. and G.L.V.; formal analysis, M.S.B., V.M.P., N.G.M., G.L.V. and H.M.E.G.; investigation, M.S.B. and V.M.P.; resources, M.S.B. and G.L.V.; data curation, M.S.B. and V.M.P.; writing—original draft preparation, M.S.B., G.L.V. and H.M.E.G. and N.G.M.; writing—review and editing, M.S.B., N.G.M., G.L.V. and H.M.E.G.; visualization, M.S.B., G.L.V. and H.M.E.G.; supervision, M.S.B.; project administration, M.S.B.; funding acquisition, M.S.B., G.L.V. and H.M.E.G. All authors have read and agreed to the published version of the manuscript.

**Funding:** This research was partially funded by Conselho Nacional de Desenvolvimento Científico e Tecnológico (CNPq), code #407463/2016-0, #311541/2021-6 and #311907/2021-0, Fundação de Amparo à Pesquisa do Estado de Mato Grosso (FAPEMAT), code #561397/2014, Programa de Grande Escala Biosfera-Atmosfera na Amazônia (LBA), Universidade Federal de Mato Grosso (UFMT), Programa de Pós-Graduação em Física Ambiental (PPGFA/IF/UFMT), Instituto Federal de Mato Grosso (IFMT), the National Science Foundation (NSF) Award Number IIA-1301346, and New Mexico State University.

**Acknowledgments:** The authors would like to thank the Distributed Active Archive Center (DAAC) of Oak Ridge National Laboratory (ORNL) and the Laboratório de Sensoriamento Remoto Aplicado à Agricultura e Floresta (LAF) of Instituto Nacional de Pesquisas Espaciais (INPE) to provide data to this research.

**Conflicts of Interest:** The authors declare no conflict of interest. The funders had no role in the design of the study; in the collection, analyses, or interpretation of data; in the writing of the manuscript, or in the decision to publish the results. Any opinions, findings, and conclusions or recommendations expressed in this material are those of the author(s) and do not necessarily reflect the views of the National Science Foundation.

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
