# Peer review of "Evapotranspiration Seasonality over Tropical Ecosystems in Mato Grosso, Brazil"

_remotesensing, doi:10.3390/rs14102482_

Round 1
Reviewer 1 Report
This manuscript offers a very important compilation of ET flux measurements in a key region of the Amazon in Brazil, and makes an important effort in comparing ground-based ET measurements with common and emerging satellite products but at the end solely compares to one (MOD16 ET). Such a dataset and primary analysis is, no doubt of interest for the readers of Remote Sensing, perhaps due to the outstanding spatial analysis that it is offered for this important region within the Amazon.
I enjoyed reading about the effort taken to compile the data set and analysis, and believe that data handling and analysis a well performed, as clearly written in the methods section.
I however have two major concerns:
- The Introduction is vague and lacks a big picture: Lines 53-57, put in context strong arguments about; i.e. Physiology and gaining knowledge on ecosystem function responding to climate change. These are indeed “big picture” items, but are not well located in the introduction and sadly strong arguments to discuss these huge themes are not offered later in the manuscript.
Of course, gaining information on Mato Grosso dynamics is of huge value, but …What is the main idea the authors would like to offer to the readers? if it is just the regional variation of the MOD16 ET product, that will be a goal. But as stated “the objectives of this study were to combine and assess existing ground- and remotely sensed ET datasets (Line 121)” and I do not see a “combined product” that will acknowledge this objective.
From the discussion it is clear that the spatial analysis is fundamental in this work, but the lack of argumentation in the Introduction on the importance of land ET measurements to perhaps a) validate of b) offer uncertainties to satellite products limits the scope of the study.
I think the authors can position their study on a larger context since they offer a state-of-the-art approach to advance knowledge related to ET variability in tropical and subtropical ecosystems, for instance some of the satellite products that they disregard in the introduction have recently been proven of value in these low latitude tropical regions (i.e. https://doi.org/10.5194/hess-15-223-2011; https://doi.org/10.5194/hess-22-1351-2018, 2018; https://doi.org/10.1016/j.jhydrol.2022.127786). Please simply argue, on how landscape contrasts in Mato Grosso offer enough spatial variation to justify that a product with a 0.05-degree spatial resolution is not useful for the goals of the study”.
- The opening statement in the discussion section is rather strong and deserves a more profound discussion since the information provided in the Results section depicts important discrepancies between the measured the remote sensed ET estimates. Again, overall the estimates and the overall study is strong, but authors are missing a great opportunity to discuss fundamental issues in the literature nowadays. Why satellites perform one way or the other on ecosystems whit contrasting land surfaces.
Through the discussion it is well stated which are the limitations of the MOD16 ET product. But I am uncomfortable on how such limitations are discussed on what it was learn with the actual ET estimates that are declared as the reference values.
Author Response
Reviewer #1
This manuscript offers a very important compilation of ET flux measurements in a key region of the Amazon in Brazil, and makes an important effort in comparing ground-based ET measurements with common and emerging satellite products but at the end solely compares to one (MOD16 ET). Such a dataset and primary analysis is, no doubt of interest for the readers of Remote Sensing, perhaps due to the outstanding spatial analysis that it is offered for this important region within the Amazon.
I enjoyed reading about the effort taken to compile the data set and analysis, and believe that data handling and analysis a well performed, as clearly written in the methods section.
I however have two major concerns:
- The Introduction is vague and lacks a big picture: Lines 53-57, put in context strong arguments about; i.e. Physiology and gaining knowledge on ecosystem function responding to climate change. These are indeed “big picture” items, but are not well located in the introduction and sadly strong arguments to discuss these huge themes are not offered later in the manuscript.
Of course, gaining information on Mato Grosso dynamics is of huge value, but …What is the main idea the authors would like to offer to the readers? if it is just the regional variation of the MOD16 ET product, that will be a goal. But as stated “the objectives of this study were to combine and assess existing ground- and remotely sensed ET datasets (Line 121)” and I do not see a “combined product” that will acknowledge this objective.
From the discussion it is clear that the spatial analysis is fundamental in this work, but the lack of argumentation in the Introduction on the importance of land ET measurements to perhaps a) validate of b) offer uncertainties to satellite products limits the scope of the study.
All of these comments point to the same criticism, that the Introduction was vague and lacked a focused objective and rationale. We added a paragraph that hopefully addresses this criticism (see lines 121-136 and 141-149).
I think the authors can position their study on a larger context since they offer a state-of-the-art approach to advance knowledge related to ET variability in tropical and subtropical ecosystems, for instance some of the satellite products that they disregard in the introduction have recently been proven of value in these low latitude tropical regions (i.e. https://doi.org/10.5194/hess-15-223-2011; https://doi.org/10.5194/hess-22-1351-2018, 2018; https://doi.org/10.1016/j.jhydrol.2022.127786). Please simply argue, on how landscape contrasts in Mato Grosso offer enough spatial variation to justify that a product with a 0.05-degree spatial resolution is not useful for the goals of the study”.
Although the vegetation in northern Mato Grosso (Amazon) is quite homogeneous, the vegetation in central and southern Mato Grosso (Cerrado and Pantanal) is quite heterogeneous. In addition, much of Mato Grosso has been deforested, which increases the heterogeneity of the state's surface. Thus, we assumed that coarse spatial resolution products such as ALEXI, SSEBop and GLEAM could introduce errors due to the loss of heterogeneity in the surface ET computation. We added some text that hopefully addresses this recomendation (see lines 69-71 and 92-95).
- The opening statement in the discussion section is rather strong and deserves a more profound discussion since the information provided in the Results section depicts important discrepancies between the measured the remote sensed ET estimates. Again, overall the estimates and the overall study is strong, but authors are missing a great opportunity to discuss fundamental issues in the literature nowadays. Why satellites perform one way or the other on ecosystems whit contrasting land surfaces.
We have added a caveat to this statement to reflect the spatial and temporal discrepancies between the measured and MOD16 ET values (see Lines 510-512). However, we did not discuss these discrepancies further in the discussion because section 4.1 already discussed the presumed reasons for these discrepancies. We think that the change in the thesis statement now better reflects the discussion in the section.
Through the discussion it is well stated which are the limitations of the MOD16 ET product. But I am uncomfortable on how such limitations are discussed on what it was learn with the actual ET estimates that are declared as the reference values.
We are not sure how to respond to this comment, as it is rather vague. Does the reviewer wish that we discussed limitations to the measured ET values?
Reviewer 2 Report
Using MOD16 ET product compared with Eddy Covariance (EC) based tower measurement in this study has been one of the popular methods among most of ET research publications, therefore I don't find the methodology is unique. According to Table 1, EC measurement availability has been varied year to year between 1 and 7 years. I understand that data won't be available all the time; however, with this type of dataset, I am not sure if data can be compared between sites to generate results such as Figures 3 & 4 and Table 2, for example.
There are several places where minor English usage corrections needed. Authors are advised to work on that.
Author Response
Our response to the reviewer’s comments are in red font below.
Reviewer #2
Using MOD16 ET product compared with Eddy Covariance (EC) based tower measurement in this study has been one of the popular methods among most of ET research publications, therefore I don't find the methodology is unique. According to Table 1, EC measurement availability has been varied year to year between 1 and 7 years. I understand that data won't be available all the time; however, with this type of dataset, I am not sure if data can be compared between sites to generate results such as Figures 3 & 4 and Table 2, for example.
The figures and table that the reviewer indicates as problematic compare measured and MOD16 values of ET during the periods when measured data were available. The fact that the time period of the measured values is not the same is not important and does not compromise the comparisons made in Figs. 3 and 4 or Table 2. The only thing that is important is that the measured values are compared to MOD16 values during the same period, which is what was done here.
There are several places where minor English usage corrections needed. Authors are advised to work on that.
Yes, thank you! We have fixed some typos throughout the text.
Round 2
Reviewer 1 Report
It appears that the authors addressed previous comments in a reasonable way.
I am just still uncomfortable with the statements between lines 53 and 57, since this study does not provide a clear discussion to these strong arguments later in the manuscript and the way the statement is presented I believe it competes with the true objectives of the overall work. I suggested removing these lines.
Instead, why not arguing about the relevance time series of ET from remote sensing products in tropical ecosystems to amounts evidence about the potential landscapes controls on ET in this low latitude regions? This will certainly position this study in a broader context. A simple exploration on recent studies where MOD16 ET has been used to inquire about variation of ET across tropical landscapes may provide hints on this.
With these suggestions the manuscript is perhaps ready for publication in remote sensing.
Author Response
Our response to the reviewer’s comments are in blue font below.
Reviewer #1
It appears that the authors addressed previous comments in a reasonable way.
The authors would like to thank the reviewer the comment about the provided responses on the revised manuscript.
I am just still uncomfortable with the statements between lines 53 and 57, since this study does not provide a clear discussion to these strong arguments later in the manuscript and the way the statement is presented I believe it competes with the true objectives of the overall work. I suggested removing these lines.
Instead, why not arguing about the relevance time series of ET from remote sensing products in tropical ecosystems to amounts evidence about the potential landscapes controls on ET in this low latitude regions? This will certainly position this study in a broader context. A simple exploration on recent studies where MOD16 ET has been used to inquire about variation of ET across tropical landscapes may provide hints on this.
The authors are grateful for the suggestion to remove the sentences between lines 53 and 57. We have replaced these lines with a statement about the potential controls of landscape on ET and how models such MOD16 ET perform over contrasting land cover in lower latitude regions (lines 53 to 56).
We understand that this suggestion has already been addressed in the last review between lines 122 and 137. We discussed the control mechanisms (rainfall, landcover and flooding) of evapotranspiration in the three biomes of Mato Grosso.
With these suggestions the manuscript is perhaps ready for publication in remote sensing.
We thank the review and the suggestions made, which clarified the manuscript.